# Influence of Alloy Substrate Treatment on Microstructure and Surface Performances of Arc-Ion Plated Gold-Like Film

**DOI:** 10.3390/ma12010180

**Published:** 2019-01-07

**Authors:** Qinghai Yu, Jingxuan Pei, Jiankang Huang, Yuejuan Zhang, Di Ma

**Affiliations:** School of Materials Science and Technology, Beijing Key Laboratory of Materials Utilization of Nonmetallic Minerals and Solid Wastes, National Laboratory of Mineral Materials, China University of Geosciences, Beijing 100083, China; peijingxuan77@163.com (J.P.); 2103170040@cugb.edu.cn (J.H.); 1003154102@cugb.edu.cn (Y.Z.); 1003151302@cugb.edu.cn (D.M.)

**Keywords:** TiN, ZrN, arc ion plating, surface performance

## Abstract

Three typical surface pretreatment strategies (grind, drawing, polishing) are employed to explore the influence of alloy substrate treatment on microstructure and surface performances of arc-ion plated TiN and ZrN films. The luminance and color of the films are measured by the color coordinate value of CIELab system (a color system which is defined by the International Commission on illumination). The crystal phases, morphology and microstructure are characterized and analyzed by scanning electron microscope (SEM) and X-ray powder diffraction (XRD). In addition, the anti-alkali, salt corrosion and anti-oxidation performances of films are systematically researched. The results show that the films with grinding pretreatment are more like gold color, “L” values are 77.27 cd/m^2^ and 80.30 cd/m^2^. The “b” value of TiN film is 29.96, which is the same as that of pure gold. The “a” value of ZrN film is 0.31, which is the same as pure silver. The density of TiN and ZrN films is the best, and both TiN and ZrN films were crystalline. They have the best anti-alkali and anti-oxidation performance. The films with drawing pretreatment show slant red color and have medium brightness values (74.07–76.37 cd/m^2^), worse compactness, obvious furrows and holes in their microstructures and worse salt corrosion and anti-oxidation performances. However, the TiN films are in amorphous states. The films with polishing pretreatment have the lowest brightness (72.66 cd/m^2^), gold-like color, superior compactness and best salt corrosion performance, which have a small number of holes. The TiN films with polishing pretreatment are also in amorphous state. Above all, alloy substrate pretreatment by grinding has the best gold-like color, brightness, compactness and corrosion resistance performance. This work exclusively sheds new light on surface pretreatment of alloy substrate by arc-ion plated films and also provides a reference for corrosion resistance performance of gold-like films.

## 1. Introduction

Vacuum arc-ion plating (AIP) gold-like film can make the stainless-steel alloy have its required ornamental value and operational performances [1,2]. Stainless steel alloy ornaments are favored by the market for their low cost and wide applicability. Alloy materials cannot be directly used in ornaments for two reasons: (1) the color itself is dull [3]; and (2) contacting with air and other surfaces lets the alloy readily oxidize and corrode, and makes the alloy quality deteriorate [4]. The surface film is considered as an ideal technique to isolate the alloy from the external environment and improve its performance and ornamental value [5,6].

Electroplating is widely used due to its simple operation and low cost. Traditional electroplating has to use some corrosive chemicals during surface treatment [7,8,9,10]. The toxic gases may be released in the coating process, and induce serious harm to the environment and operators’ bodies [11]. Vacuum AIP is an advanced and eco-friendly coating method [12,13]. Under vacuum conditions, the target is ionized by arc discharge, and the method has two characters of good ionization rate and high substrate-to-film adhesion strength. Ion-plated titanium nitride and zirconium nitride are two kinds of gold-like films, and have attracted much attention [14,15,16]. The surface treatment method has a close relationship with the film structure and performance, but it is rare to see a systematic study of the influence of the surface treatment method on the structure and surface properties of the two films: (1) how the surface treatment method affects the color brightness and color of the film; (2) how to affect the surface and microstructure of films; and (3) how to affect the performance of the film.

To search for a suitable eco-friendly coating method for the ornaments, this work employs three surface pretreatment methods, and deposits titanium and zirconium nitride films on a stainless-steel alloy in AIP system. The influence of surface treatment method on the structure and anti-alkali, salt corrosion and anti-oxidation performance is systematically investigated.

## 2. Experimental

### 2.1. Materials and Instruments

Sodium hydroxide (NaOH), distilled water, 99.99% titanium (Ti), 99.99% zirconium (Zr), and stainless-steel alloy (304) were supplied from Beijing Chemical Workstation (Beijing, China). The multi-AIP system was obtained from Shili Yuan Beijing Co., Ltd. (Beijing, China). The sandpaper 2000# and cloth wheel were purchased in Beijing by a commercial route.

### 2.2. Characterization

The CIELab of the samples was assessed by portable spectrophotometer CM2600d (Konica Minolta, Tokyo, Japan). The light source is D65 and the standard observation angle is CIE10°. The phase structure of samples was analyzed with the X-ray diffractometer (XRD; Model XD-3, Rigaku, Tokyo, Japan) using CuKα radiation. All XRD patterns were obtained from 5° to 80°. The surface topography and morphology of samples were researched by a field-emission scanning electron microscope (SEM, JEOL, Tokyo, Japan).

### 2.3. The Preparation of Gold-Like Film

Three stainless-steel alloy coupons with dimensions of 20 mm × 20 mm × 1.5 mm were used as the substrate. The smooth surface served as a comparison, whereas the surface drawing and polishing were carried out by sandpaper 2000# and a cloth wheel, respectively. The substrate was ultrasonically cleaned in an acetone solution and heated to 50 °C in a water bath for 15 min to remove the surface oil. Then, the substrate was placed in an absolute ethanol solution and ultrasonically cleaned, followed by heating at 50 °C in a water bath to remove the residual acetone. Afterwards, the substrate was placed in distilled water and shook in an ultrasonic cleaner to remove positive and negative ions from the surface. Finally, the surface water was blown off with clean nitrogen gas, and the substrate was put into the AIP system ready for deposition. The main deposition parameters include: heating temperature (100 °C), base vacuum (2.0 × 10^−3^ Pa), glow discharge cleaning (−200 V substrate bias with a duty cycle of 50% for 6 min), deposition vacuum of 1.0 × 10^−1^ Pa, bias voltage of −60 V and target current of 80 A. Table 1 lists surface treatments of samples deposited by the AIP system. Six samples were involved with three surface treatments and the two films. The three surface treatments on the substrate were smooth surface, drawing surface and polished surface; and were named as −1, −2 and −3, respectively. The two films were titanium nitride (TiN) and zirconium nitride (ZrN), and the thickness of the two films is about 0.2 μm. Hereafter, the samples were marked as 1, 2, 3, 4, 5 and 6 accordingly.

### 2.4. The Surface Treatment of Gold-Like Film

Firstly, 50 mL solution of 36% NaOH was selected as the alkali solution and the experimental period was 36 h. The coated samples were washed with alcohol under ultrasonic for 20 min, then dried and were put into NaOH solution staying for 36 h. Afterwards, the samples were washed by ultrasonic for 10 min in alcohol, and the corrosion morphology was observed by SEM after drying to investigate the corrosion resistance of the film. Then, 3 g/L NaCl solution (human sweating concentration) was employed to test the corrosion resistance in a salty environment for 24 h. After that, all the samples were rinsed with distilled water and dried at room temperature. The anti-oxidation performance was researched by drying samples at 100 °C for 24 h.

## 3. Results and Discussion

### 3.1. CIELab Color Coordinate Value Test

Based on physiological characteristics, the CIELab color value is a device-independent color model [17]. It consists of three dimensionless element values of “L”, “a”, “b”. The “L” (Luminosity) value reflects the intensity of the surface luminescence of the illuminant; “a” and “b” are two color values, and the “a” value reflects the color from dark green to pink; the “b” value represents the color from bright blue to yellow.

Figure 1 shows the CIELab color values of six samples. As shown in Figure 1, we can see the effect of the surface treatments and films on the samples’ ornamental performance, including the brightness and the color. As for the surface treatment method, we can see that the “L” values of the samples 1 and 4 are high from the “L” value curve in Figure 1, indicating that the ordinary smooth processed samples have the highest brightness value and the highest reflection intensity. This may be due to the increase in the reflective surface and the loss of light after the drawing and polishing process [18]. Sample 4 (ZrN film with grind pretreatment) has an “L” value of 80.3, which is closest to the “L” value of gold (88–89 cd/m^2^) and silver (86–89 cd/m^2^). It can be seen from the “a” and “b” values in Figure 1. The results of the three surface treatments are not very different, indicating that the surface treatment has little effect on the surface color of the coated samples. Sample 1 (TiN film with grind pretreatment) has a “b” value of 29.96, which is almost identical to the “b” value of pure gold (30–35). In summary, the “L”, “a”, and “b” values of the TiN film are close to pure gold (the b value of gold is 30–35); the “a” value of the ZrN film is consistent with pure silver (−6~4), and the “L” value and the “b” value are close to the pure silver (the b value of silver is 0~2). It shows a bright white color. In terms of the ornamental performance of gold and silver, it can be seen that the rank list for the luminosity of the three surface pretreatments is: grind pretreatment > drawing pretreatment > polishing pretreatment. The color of TiN film is close to pure gold, while the color of ZrN film is close to pure silver.

### 3.2. Crystal Structure and Surface Morphology

Figure 2 shows the XRD spectra of the TiN film on the surface of the substrate under three surface pretreatment methods. It can be seen from Figure 2 that the TiN film exhibited various crystal orientations, and the surface pretreatment method of the substrate directly affected the crystal structure and preferred orientation of the film crystal. On the smooth surface (TiN-1), the diffraction peaks corresponding to the (111), (200) and (311) crystal faces of the TiN crystal were clearly observed. The diffraction peak of the (111) crystal plane was gentle and had the lowest intensity, indicating that a small amount of amorphous TiN was deposited on the surface of the substrate. The diffraction peaks of the (200) and (311) crystal planes were sharp and the crystals were in good condition. Among them, the (311) crystal plane had the highest diffraction peak intensity, and the crystal grains had preferential growth along the (311) crystal plane. On the drawing surface (TiN-2), the diffraction peak of the (111) crystal plane disappeared, the intensity of the diffraction peak of the (311) crystal plane was greatly reduced, and the peak position of the (200) crystal plane diffraction peak was slightly shifted to the left, and the peak shape was gentle. This indicated that, after the surface drawing treatment, the amorphous TiN on the (111) crystal plane of the substrate disappeared, and the preferential growth of the crystal grains changed from the (311) crystal plane to the (200) crystal plane. The film consisted essentially of amorphous TiN grown along the (200) crystal plane. Polished surface (TiN-3), similar to TiN-2, only the diffraction peaks of the two crystal faces of TiN grains (200) and (311) were observed, and the grains were preferentially grown on the (200) crystal plane, and the film was mainly non-crystalline TiN [19]. After the surface treatment of the substrate by the above three methods, the surface roughness was different, and the surface state was also significantly different. In the process of depositing thin films by ion plating, TiN grains selected different nucleation sites on the surface of three kinds of substrates, which lead to the preferential growth of crystal grains [20]. The results showed that the TiN film deposited on the surface of the polished surface substrate was a crystalline film by ion plating technology, and the crystal grains grew preferentially along the (311) crystal plane. The TiN film deposited on the substrate of the drawing surface and the polishing surface was mostly an amorphous film, and there was a preferred orientation at the (200) crystal face.

Figure 3 shows the XRD spectra of the ZrN film on the surface of the substrate under three surface pretreatment methods. It can be seen from Figure 4 that the ZrN film had various crystal plane orientations, and the surface pretreatment method of the substrate directly affected the preferred orientation of the film crystal. On the smooth surface (ZrN-1), the (220) crystal plane had the highest diffraction peak intensity and the ZrN crystal grains grew preferentially [21]. The diffraction peaks of the (111), (200), and (311) crystal faces were slightly lower, and the crystal grains were good. On the drawing surface (ZrN-2), the diffraction peak intensity of each crystal plane was lower than that of the smooth surface, and the crystal grains showed no obvious preferential growth. Polished surface (ZrN-3), the diffraction peak intensity of each crystal plane was the strongest, and the crystal grains grew preferentially along the (111) and (220) crystal planes, and the film was mainly crystalline ZrN. The results showed that crystalline ZrN was deposited on the surface of the substrate after three kinds of surface treatments. However, the surface treatment method directly affected the grain growth condition: the preferred orientation of the grains did not appear in the film on the drawing surface, but the film on the smooth surface and polished surface had preferential growth of crystal grains. This may be because the surface roughness of the brushed substrate was too large, and the surface energy was too high, which was not conducive to the nucleation and growth of ZrN crystals [22].

Figure 4 shows the SEM images of TiN and ZrN films on the surface of the substrate under three surface treatment methods. Due to the different surface and different corrosion treatments of the films, the different plotting scales are used in SEM images for a favorable observation. As can be seen from Figure 4, the deposited film on the substrate of the smooth surface and the polished surface was more continuous and dense; the film denseness on the drawing surface was deteriorated. For the TiN film, the smooth surface (Figure 4a) had the most compact film and no obvious corrosion sites, which helps to improve the corrosion resistance of the substrate. The film on the drawing surface in Figure 4b could clearly observe the furrow left by the wire drawing treatment, and a small amount of TiN particles were present. The surface of the substrate was directly exposed after the TiN particles were peeled off, which may cause corrosion of the substrate. The film on the polished surface (Figure 4c) had a small number of holes, which reduced the compactness of the film and provided sites for pitting corrosion, which was not conducive to improving the corrosion resistance of the substrate [23]. For the ZrN film, the films on the smooth surface (Figure 4d) and polished surface (Figure 4f) were dense, but there was a small number of holes. The film on the drawing surface (Figure 4e) not only had a large number of furrows, but also had many holes in the film and poor compactness. Compared to the drawing surface, the film on the smooth surface and the polished surface were more advantageous for improving the corrosion resistance of the substrate. Figure 4 also shows the pictures of stainless-steel alloy (g), stainless-steel alloy with TiN coating film (h) and stainless-steel alloy with ZrN coating film (i) for a comparison.

### 3.3. Surface Performances

Figure 5 shows the SEM images of the salt-treated samples. As shown in Figure 5, we can see the effect of the films and surface treatments on the salt corrosion resistance of the samples. As for the surface treatments, Sample 1 (grind surface) (Figure 5a,d) has slight corrosion spots and marks, and indicates that its salt corrosion resistance is in a middle level. Sample 2 (drawing surface) (Figure 5b,e) has obvious corrosion stains and white spots, and indicates that its salt corrosion resistance is poor [24]. The reason may be that the wire drawing treatment increases the corrosion area of the film layer, and makes the corrosion readily extend. Sample 3 (polished surface) (Figure 5c,f) has no obvious corrosion marks, and its coating is complete. This means that its salt corrosion resistance is admirable. As for the films, observing and comparing the surface morphologies in Figure 5 shows that the corrosion level of the TiN films is superior to that of ZrN films under the same treatments. This indicates that the salt corrosion resistance of the TiN films is better than that of the ZrN films in majority. In this way, the rank list for the salt corrosion resistance of the three surface treatments is: polishing > grind > drawing; and the resistance of the TiN film is better than that of the ZrN film.

Figure 6 shows the SEM images of the samples after alkali etching, As shown in Figure 6, we can see the effect of films and surface treatments on the alkali corrosion resistance of the samples. As for the surface treatments, Sample 1 (smooth surface) (Figure 6a,d) shows no obvious corrosion marks and the films’ layer is intact, and indicates that its alkali corrosion resistance is admirable. Sample 2 (drawing surface) (Figure 6b,e) has slight corrosion marks and a few layers fall off, and indicates that its alkali corrosion resistance is poor [25]. The reason may be that the drawing surface increases the corrosion area of the film layer and makes the corrosion easier; Sample 3 (polished surface) (Figure 6c,f) is slightly corroded with white spots and small pits, and indicates that its alkali corrosion resistance is in a middle level. As for the films, observing and comparing the surface morphologies in Figure 6 shows that the corrosion level of the TiN films is superior to that of ZrN films under the same treatments, and indicates that the alkali corrosion resistance of the TiN films is better than that of the ZrN films in majority. In this way, the rank list for the alkali corrosion resistance of the three surface treatments is: smooth > polishing > drawing; the alkali corrosion resistance of TiN film is higher than that of ZrN film.

Figure 7 shows the SEM images of the samples after oxidation, As shown in Figure 7, we can see the effect of films and surface treatments on the oxidation resistance of the samples. As for the surface treatments, Sample 1 (smooth surface) (Figure 7a,d) has no corrosion, and indicates that its oxidation resistance is admirable. Sample 2 (drawing surface) (Figure 7b,e) shows obvious cracks on the surface and the film layers are peeled off, and indicates that its oxidation resistance is poor. In the case of Sample 3 (polished surface) (Figure 7c,f), the TiN film is severely corroded, and the film layer fell off to form a small pit; the ZrN film is non-corrosive, and indicates that its oxidation resistance is in a middle level [26]. As for the films, observing and comparing the surface morphologies in Figure 7 shows that the corrosion level of the ZrN films is superior to that of TiN films under the same treatments, and indicates that the oxidation resistance of the ZrN films is better than that of the TiN films in majority. In this way, the rank list for the oxidation resistance of the three surface treatments is: smooth > polishing > drawing; the oxidation resistance of the ZrN film is higher than that of the TiN film.

It can be seen from the above analysis that the TiN film with a smooth surface can impart the best comprehensive corrosion resistance to the stainless steel 304 alloy. The rank list for comprehensive corrosion resistance of the three surface treatments is: smooth > polished > drawing; the salt and alkali resistance of the TiN film is better than that of the ZrN film, but the oxidation resistance of the TiN film is lower than that of the ZrN film.

## 4. Conclusions

The vacuum arc-ion plating gold-like film method is the optimal strategy to embellish and protect stainless-steel alloy ornaments. The application of typical gold-like films (TiN, ZrN) is limited by the lack of research about the influence of substrate surface treatment methods on the films’ structure and properties. In this work, the surface of the alloy was pretreated by three typical methods, including grind, drawing and polishing, and was deposited with TiN and ZrN films by arc-ion plating method. This research systematically investigated the effects of alloy surface treatment methods on film brightness, color, morphology and structure, as well as corrosion resistance and oxidation resistance. The main conclusions are as follows:(1)The films with grinding pretreatment are more like gold color; the “L” values are 77.27 cd/m^2^ and 80.30 cd/m^2^. The “b” value of TiN film is 29.96, which is the same as that of pure gold. The “a” value of ZrN film is 0.31, which is the same as pure silver. The density of TiN and ZrN films are the best, and both TiN and ZrN films were crystalline. They have the best anti-alkali and anti-oxidation performance. The films with polishing pretreatment have the lowest brightness (72.66 cd/m^2^) and gold-like color. In terms of the ornamental performance of gold and silver, the rank list for the luminosity of the three surface treatments is: ordinary smoothing treatment > drawing treatment > polishing treatment; TiN film color is closer to pure gold, and ZrN film is closer to pure silver.(2)The surface treatment method affects the compactness of the film layer and the growth state of the crystal grains. The film obtained by drawing has poor compactness and obvious furrows and cavities; the film obtained by grinding and polishing has good compactness, which is beneficial to improve the corrosion resistance of the film. From grind treatment to drawing and polishing treatment process, the TiN film translates from crystalline to amorphous, and the preferred orientation changes from (311) crystal plane to (200) crystal plane; ZrN film has a good crystallinity, and the preferred orientation along (220) crystal plane appears after disappearing first.(3)The corrosion resistance and oxidation resistance of the gold-like film are affected by the surface treatment method and film type. The film obtained by the grinding treatment has the best alkali resistance and oxidation resistance. The film obtained by the polishing treatment has the best salt corrosion resistance. The gold-like film obtained by the drawing process has the worst corrosion resistance and oxidation resistance because of a large number of furrows and cavities. TiN film has better corrosion resistance and worse oxidation than ZrN film.

Therefore, the alloy substrate treated by the grinding process is advantageous for obtaining a gold-like film with a high brightness value and a gold-like color. The film has high compactness, good crystal crystallization, and optimal corrosion resistance and anti-oxidation performances. It is hopeful to meet the decoration and protection needs of alloy ornaments by depositing a gold-like film on the surface of a polished stainless-steel substrate with the AIP method.

## Figures and Tables

**Figure 1 materials-12-00180-f001:**
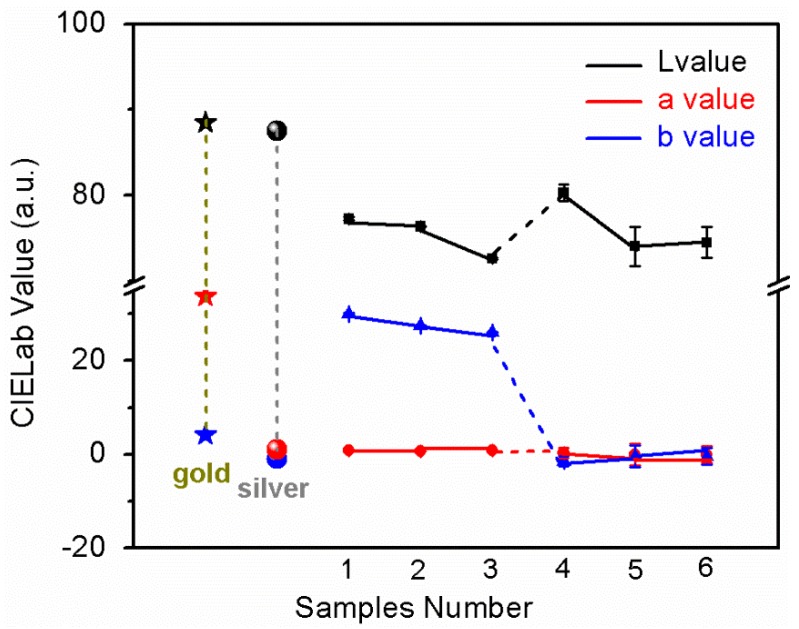
The L, a, and b value of gold, silver and prepared samples.

**Figure 2 materials-12-00180-f002:**
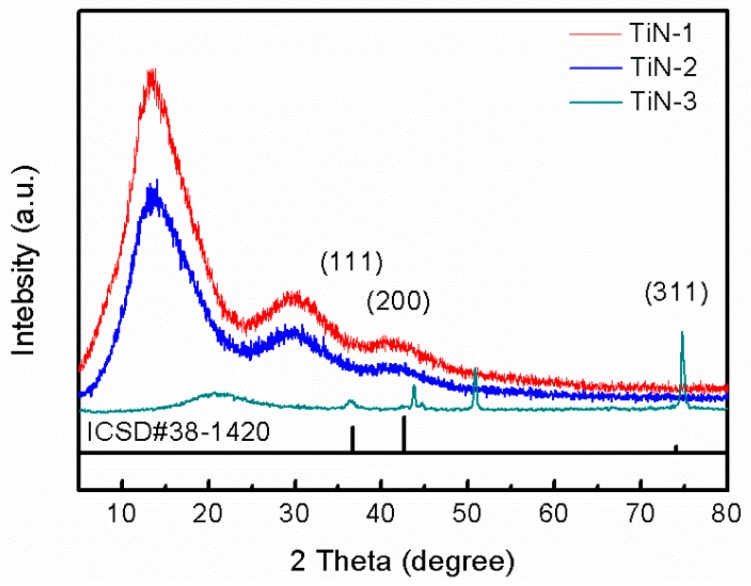
The X-ray powder diffraction (XRD) patterns of the TiN films deposited at different substrates: TiN-1 corresponds to the smooth substrate, TiN-2 represents the drawing surface substrate and TiN-3 refers to the polished substrate.

**Figure 3 materials-12-00180-f003:**
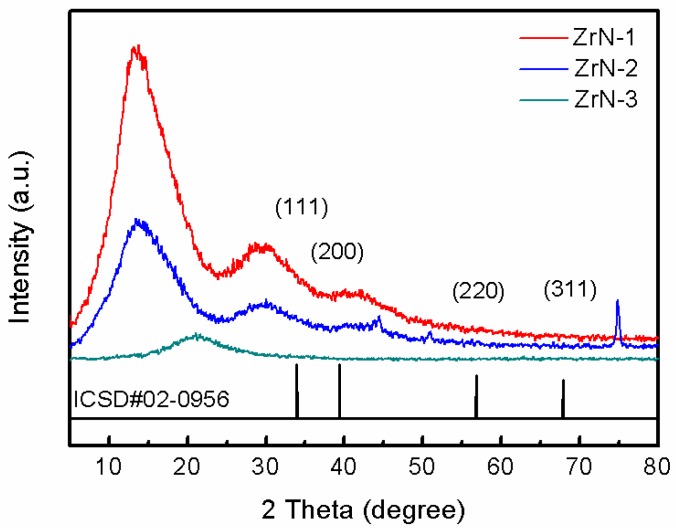
The XRD patterns of the ZrN films deposited at different substrates: ZrN-1 corresponds to the smooth substrate, ZrN-2 represents the drawing surface substrate and ZrN-3 refers to the polished substrate.

**Figure 4 materials-12-00180-f004:**
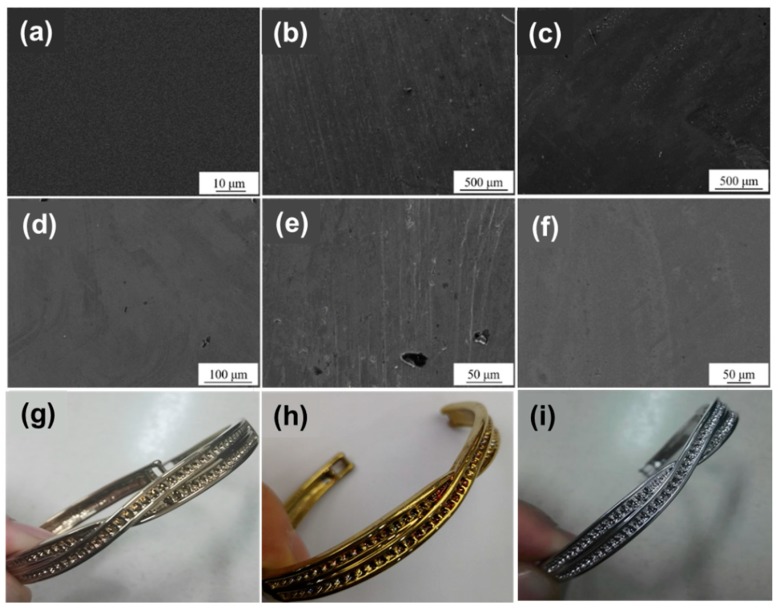
The scanning electron microscope (SEM) images of the as-prepared samples: TiN-1 (**a**), TiN-2 (**b**), TiN-3 (**c**), ZrN-1 (**d**), ZrN-2 (**e**), ZrN-3 (**f**); the picture of (**g**) stainless-steel alloy, (**h**) TiN, (**i**) ZrN.

**Figure 5 materials-12-00180-f005:**
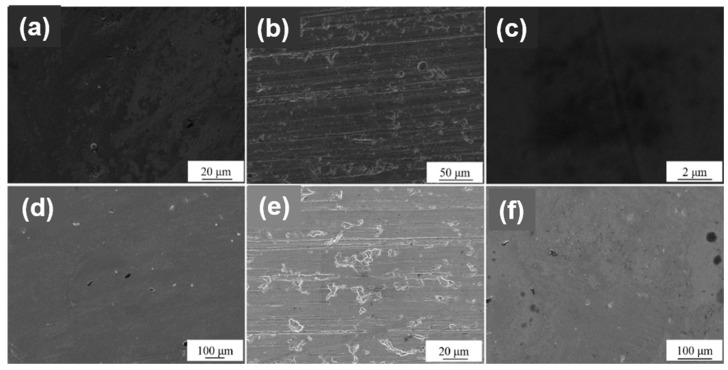
The SEM images of the salt-treated samples: TiN-1 (**a**), TiN-2 (**b**), TiN-3 (**c**), ZrN-1 (**d**), ZrN-2 (**e**), ZrN-3 (**f**).

**Figure 6 materials-12-00180-f006:**
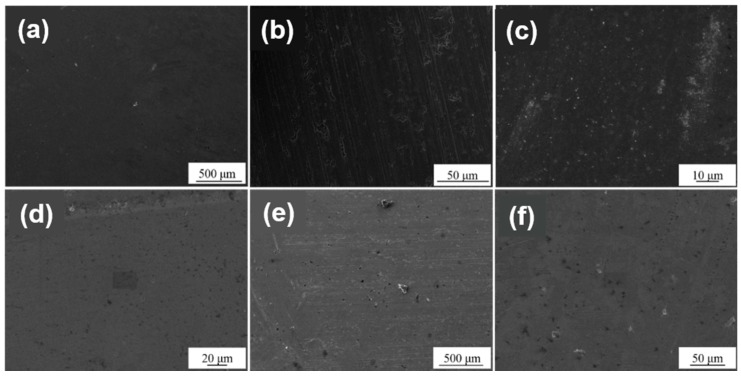
The SEM images of the alkaline-treated samples: TiN-1 (**a**), TiN-2 (**b**), TiN-3 (**c**), ZrN-1 (**d**), ZrN-2 (**e**), ZrN-3 (**f**).

**Figure 7 materials-12-00180-f007:**
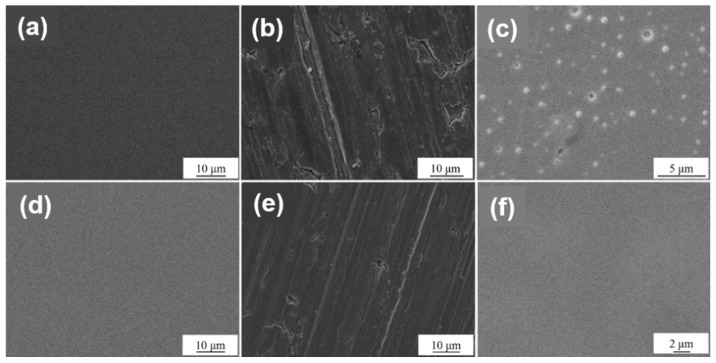
The SEM images of oxidized samples: TiN-1 (**a**), TiN-2 (**b**), TiN-3 (**c**), ZrN-1 (**d**), ZrN-2 (**e**), ZrN-3 (**f**).

**Table 1 materials-12-00180-t001:** Surface treatments for samples deposited by the arc ion plating (AIP) system.

Numbers	Samples	Surface Treatments	Films
1	TiN-1	Grinding surface	TiN film
2	TiN-2	Drawing surface	TiN film
3	TiN-3	Polished surface	TiN film
4	ZrN-1	Grinding surface	ZrN film
5	ZrN-2	Drawing surface	ZrN film
6	ZrN-3	Polished surface	ZrN film

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
