# Peer review of "Influence of Alloy Substrate Treatment on Microstructure and Surface Performances of Arc-Ion Plated Gold-Like Film"

_materials, 2019, doi:10.3390/ma12010180_

Reviewer 1 Report

The manuscript written by the authors are interesting but looks that there is no novelty. However, it has many drawbacks to be considered for publication. I have some suggestion which need to be addressed by the authors.

1.      Abstract section need some quantitative data analysis of obtained results.

2.      Experimental section is not clear thus it needs to be rewritten.

3.      In table 2, what does it mean by “a” and “b”?

4.      What is the unit of “L” in table 2?

5.      What does mean by sample number 1-6? There are no details about the number in the manuscript.

6.      What does mean of Fig. 1 a, b, c and d? There is no description of these sub-figures. What authors are trying to explain by this figure? Why authors have not explained in the text about this result?

7.      There is no reference in the result and discussion section then how authors can justify their results?

8.      There is no scientific discussion of obtained results. It lacks of scientific contribution in the manuscript.

9.      How authors can justify by doing XRD of substrate properties while coating was deposited? It is not relevant.

10.  What is the thickness and bond strength of coating deposited on different substrate?

11.  The manuscript has lack of experimental results and proper discussion of the obtained results. Thus, I concluded that this manuscript cannot be considered for publication in its present form.

Author Response

Thank you for your valuable comments on our manuscript. We have revised the manuscript carefully based on your comments. We here with provide our response to your comments as follows:

The manuscript written by the authors are interesting but looks that there is no novelty. However, it has many drawbacks to be considered for publication. I have some suggestion which need to be addressed by the authors.

Question 1: Abstract section need some quantitative data analysis of obtained results.

Author reply:

Thanks for your valuable comments. Thanks for your comment. Some quantitative data analysis and results are added in abstract section in the revised manuscript.

List of changes and modifications:

(1) The results show that the film with grinding pretreatment has a highest brightness(80.30cd/m2), a gold-like colour and a best compactness. All the films with grinding pretreatment are crystalline state and have best anti-acid and anti-alkali activities.

(2) The films with drawing pretreatment show slant red color and have medium brightness value (74.07-76.37 cd/m2), a worst compactness, obvious furrows and holes in their microstructures and worst salt corrosion and anti-oxidation performances. But the TiN films are amorphous state.

(3) The films with polishing pretreatment have a lowest brightness (72.66 cd/m2), a gold-like colour a superior compactness and a best salt corrosion performance, which have small number of holes. The TiN films with polishing pretreatment are also amorphous state.

Question 2: Experimental section is not clear thus it needs to be rewritten.

Author reply:

The experimental section has been rewritten to be clear. Thanks for the advice.

Question 3: In table 2, what does it mean by “a” and “b”?

Author reply:

Thank you for the question. The ‘a’ represents a red to green color value.  In color coordinate system, the positive ‘a’ means that the color towards red, in contrast, the negative ‘a’ value are biased toward the green. The ‘b’ represents a yellow-blue color value, which the positive b indicate that the color of the alloy film changed to yellow. We have explained them in revised manuscript.

Question 4:

What is the unit of “L” in table 2?

Author reply:

The unit of “L” is nit (cd/m2). We have added it in revised manuscript. Thanks for the question.

Question 5:

What does mean by sample number 1-6? There are no details about the number in the manuscript.

Author reply:

The sample number of 1-6 means TiN-1, TiN-2, TiN-3, ZrN-1, ZrN-2 and ZrN-3, respectively. We have given an explanation in the section of 2.3 and table 1. Thanks.

Question 6:

What does mean of Fig. 1 a, b, c and d? There is no description of these sub-figures. What authors are trying to explain by this figure? Why authors have not explained in the text about this result?

Author reply:

Thank you for your question. We have revised the Fig.1 to be clear. The Fig.1a shows the value of “L”, “a”, “b” of gold and prepared samples. And Fig.1b shows the value of “L”, “a”, “b” of silver and prepared samples. We have given a detailed explanation in revised manuscript.

Question 7:

There is no reference in the result and discussion section then how authors can justify their results?

Author reply:

We have supplemented the references in the result and discussion section. Thanks.

Question 8:

There is no scientific discussion of obtained results. It lacks of scientific contribution in the manuscript.

Author reply:

Thank you for your comment. More discussions are supplied in the part of 3.

Question 9:

How authors can justify by doing XRD of substrate properties while coating was deposited? It is not relevant.

Author reply:

The XRD can’t justify the substrate properties while coating was deposited.  We only employed it to explain the crystal phase and crystal facet which further confirmed the coating of TiN and ZrN. The standard JCPDS of TiN and ZrN are added in the XRD patterns. The diffraction peaks can be indexed to the phase of TiN and ZrN. Thanks.

Question 10:

What is the thickness and bond strength of coating deposited on different substrate?

Author reply:

The thickness of films is about 0.2μm. And from Fig. 4(g, h, i), we can find that the film has a good bond strength with the alloy substrate. We have added in the paper. Thanks.

Question 11:

The manuscript has lack of experimental results and proper discussion of the obtained results.

Author reply:

Thank you for your comment. More discussions and experimental results are supplied in the revised manuscript.

Reviewer 2 Report

The article presented by Qinghai Yu et al, regarding the influence of substrate surface on the functional properties of TiN and ZrN alloy films, although interesting to certain extend, requires a serious change on the general guideline of the article and major amendments. Please find attach to this message my comments:

The title does not convey in which aspect the “influence” of substrate is assessed. Which functional or physico/chemical aspects are investigated?  Corrosive and reflective properties? This information has to be somehow included in the title.

The literature review does not seems be comprehensive enough. Several works on TiN/ZrN are neglected and no references from MDPI are present in the article. Please expand upon references works and include them in the introduction: I.e: 10.1016/j.matchar.2017.10.016, 10.1002/sia.6371, 10.1134/S1063785018010224, 10.1016/j.jallcom.2018.07.090,10.1016/j.actamat.2017.03.053.

Since the chromatic properties of the film are at the core of the document, the reviewer would suggest to include images/pictures of the actual surfaces of the samples.

There is no information of thicknes or the coatings, the chromatic aspects can be strongly influenced by sub micrometric thicknesses comparable with both visible light wavelength and roughness of the substrates

XRD patterns are not indexed and several peaks are  (TIN-1) are not indexed. The shifting  of 5 deg in 2thetha is higly unlikely. Nor even under high strain.

The corrosion experiments are not standard. The alkaline and acidic tests are performed without much care or attention to standard procedures. Corrosion without TAFEL plots is definitely not recommended. If the authors prefer to show corrosion effects with SEM images, Cross sections and elemental mappings EDX are needed in order to evaluate degradation and failure of the coatings.

Author Response

Thank you for your valuable comments on our manuscript. We have revised the manuscript carefully based on your comments. We here with provide our response to your comments as follows:

The article presented by Qinghai Yu et al, regarding the influence of substrate surface on the functional properties of TiN and ZrN alloy films, although interesting to certain extend, requires a serious change on the general guideline of the article and major amendments.

Question 1: 

The title does not convey in which aspect the “influence” of substrate is assessed. Which functional or physico/chemical aspects are investigated?  Corrosive and reflective properties? This information has to be somehow included in the title.

Author reply:

Thank you for your advice. The anti-acid, alkali, salt corrosion and anti-oxidation performances of the gold-like film are investigated. We collectively call them as surface performances. We have revised the title asInfluence of alloy substrate treatment on microstructure and surface performances of arc-ion plated gold-like film.

Question 2: 

The literature review does not seems be comprehensive enough. Several works on TiN/ZrN are neglected and no references from MDPI are present in the article. Please expand upon references works and include them in the introduction: I.e: 10.1016/j.matchar.2017.10.016, 10.1002/sia.6371, 10.1134/S1063785018010224, 10.1016/j.jallcom.2018.07.090,10.1016/j.actamat.2017.03.053.

Author reply:

Thank you for your suggestion. The relevant references are added in revised manuscript.

Question 3: 

Since the chromatic properties of the film are at the core of the document, the reviewer would suggest to include images/pictures of the actual surfaces of the samples.

Author reply:

Thank you for your comment. The images/pictures of the actual surfaces of the samples are added in Fig.4 in revised manuscript.

Question 4: 

There is no information of thickness or the coatings, the chromatic aspects can be strongly influenced by sub micrometric thicknesses comparable with both visible light wavelength and roughness of the substrates.

Author reply:

Thank you for your doubt. The thickness of films is about 0.2 μm. And the pictures of the actual surfaces of the samples are added in Fig. 4. The color of the substrates has been covered by TiN and ZrN films completely.

Question 5: 

XRD patterns are not indexed and several peaks are (TIN-1) are not indexed. The shifting of 5 deg in 2thetha is higly unlikely. Nor even under high strain.

Author reply:

Thank you for your comment. We have added the standard JCPDS of TiN and ZrN in revised manuscript. The diffraction peaks can be indexed to the phase of TiN and ZrN. The main diffraction peaks are in the same degree, which don’t have a large shifting.

Question 6: 

The corrosion experiments are not standard. The alkaline and acidic tests are performed without much care or attention to standard procedures. Corrosion without TAFEL plots is definitely not recommended. If the authors prefer to show corrosion effects with SEM images, Cross sections and elemental mappings EDX are needed in order to evaluate degradation and failure of the coatings.

Author reply:

Many thanks for your comment. The standard procedures are added in the “Experimental” section in our revised manuscript. TAFEL test is a very important method in electrochemical corrosion field. However, in our experiments, the SEM is the convenient and direct method to investigate the chemical corrosion morphology of the TiN and ZrN films. And we have added the detailed explanation of the SEM pictures to illuminate the corrosion performance. Cross sections and elemental mappings EDX will be conducted in our future research.

Reviewer 3 Report

 “CIELab” is recommended to be defined at the first time used in the manuscript, for instance in line 49.

Would be great if you add some Refs. In line 27, and also other statements that are not the authors’ own statements.

I would suggest the authors to elaborate the main research question of the work in the last paragraph of the introduction. It is not very clear at the moment.

“were bought from” in line 54 can be changed to “were supplied from” or ” were sourced from”

An introduction to Table 1 is necessary before the table. It is unclear right now.

An explanation about “Drawing surface” could add value.

An introduction to Table 2 is necessary. The readers do not understand what are the parameters given in the table.

Would be great if there would be an explanation about the “Cielab Color Coordinate Value Test” in the “experimental procedure” part

Error bars in the graphs are highly recommended.

What is the thickness of the films? The three peaks shown in Fig. 2 for TiN-1 are for the stainless steel substrate and not the film, since the file is rather thin. TiN-2 and TiN-3 are also following the same trend without any significant difference. The difference in the intensity can be seen from initial degrees to the last degrees, so I am not sure if it can be attributed to the effect of the substrate roughness.

Can the authors explain why the XRD peaks in TiN-1 and ZrN-1 are different from the others?

Could the authors justify how that would be possible to judge about the corrosion performance of the coatings only through surface topography evaluation? The explanation about the corrosion performance of the films are not enough.

Author Response

Thank you for your valuable comments on our manuscript. We have revised the manuscript carefully based on your comments. We here with provide our response to your comments as follows:

Question 1

“CIELab” is recommended to be defined at the first time used in the manuscript, for instance in line 49.

Author reply:

Thank you for your comment. We have revised the introduction, so we explain the CIELab at the first time at the part of 2.2 in revised manuscript.

List of changes and modifications:

The CIELab (a color system which defined by International Commission on illumination) of the samples was assessed by portable spectrophotometer CM2600d (Konica Minolta, Tokyo, Japan). The light source is D65 and the standard observation angle is CIE10o

Question 2

Would be great if you add some Refs. In line 27, and also other statements that are not the authors’ own statements.

Author reply:

The references are added in line 27. Thank you for your advice.

Question 3

I would suggest the authors to elaborate the main research question of the work in the last paragraph of the introduction. It is not very clear at the moment.

Author reply:

We have rewritten the introduction part to elaborate the main research of our work. Thank you.

Question 4

were bought from” in line 54 can be changed to “were supplied from” or were sourced from”

Author reply:

We are sorry for the slip. We have changed the were bought from” to “were supplied from”. Thanks.

Question 5

An introduction to Table 1 is necessary before the table. It is unclear right now.

Author reply:

Thank you for the advice. An introduction was supplied before the table 1 to explain it.

List of changes and modifications:

Table 1 lists surface treatments of samples deposited by AIP system. Six samples were involved with three surface treatments and the two films. The three surface treatments on the substrate were smooth surface, drawing surface and polished surface; and were named as -1, -2 and -3, respectively. The two films were titanium nitride (TiN) and zirconium nitride (ZrN). Hereafter, the samples were marked as 1, 2, 3, 4, 5 and 6 accordingly.

Question 6

An explanation about “Drawing surface” could add value.

Author reply:

Thank you for your comment. The surface treatment method has a close relationship with the film structure and performances, but it is rare to see a systematic study of the influence of the surface treatment method on the structure and surface properties of the two films: (1) how the surface treatment method affects the color brightness and color of the film; (2) how to affect the surface and microstructure of films; and (3) how to affect the performance of the film.

Question 7

An introduction to Table 2 is necessary. The readers do not understand what are the parameters given in the table.

Author reply:

Thank you for the important suggestion.  The ‘a’ represents a red to green color value. In color coordinate system, the positive ‘a’ means that the color towards red, in contrast, the negative ‘a’ value are biased toward the green. The ‘b’ represents a yellow-blue color value, which the positive b indicate that the color of the alloy film changed to yellow. We have given an explanation in revised manuscript.

Question 8

Would be great if there would be an explanation about the “Cielab Color Coordinate Value Test” in the “experimental procedure” part.

Author reply:

Thank you for your suggestion. We have added the explanation about the “Cielab Color Coordinate Value Test” in the “experimental” part.

List of changes and modifications:

The CIELab (a color system which defined by International Commission on illumination) of the samples was assessed by portable spectrophotometer CM2600d (Konica Minolta, Tokyo, Japan). The light source is D65 and the standard observation angle is CIE10o

Question 9

Error bars in the graphs are highly recommended.

Author reply:

Special thanks for your comment. We did experiments for several times. We have added the error bars in our experiments.

Question 10

What is the thickness of the films? The three peaks shown in Fig. 2 for TiN-1 are for the stainless steel substrate and not the film, since the file is rather thin. TiN-2 and TiN-3 are also following the same trend without any significant difference. The difference in the intensity can be seen from initial degrees to the last degrees, so I am not sure if it can be attributed to the effect of the substrate roughness.

Author reply:

Thank you for your question. The thickness of films is about 0.2 μm. In XRD patterns, the main diffraction peak belongs to stainless-steel not the film. But the other diffraction peaks can be indexed to the phase of TiN and ZrN. We have added the standard JCPDS of TiN and ZrN in the revised manuscript. Thanks.

Question 11

Can the authors explain why the XRD peaks in TiN-1 and ZrN-1 are different from the others?

Author reply:

Thank you for your comment. We have added the standard JCPDS of TiN and ZrN in revised manuscript. The diffraction peaks can be indexed to the phase of TiN and ZrN. The main diffraction peaks are in the same degree, which don’t have a large shifting. So, it can be explained the phase of TiN-1 and ZrN-1.

Question 12

Could the authors justify how that would be possible to judge about the corrosion performance of the coatings only through surface topography evaluation? The explanation about the corrosion performance of the films are not enough.

Author reply:

Thank you for your comment. The explanation of corrosion performance has been added and revised in the new manuscript. In our experiments, the SEM is the intuitive way to investigate the corrosion morphology of the TiN and ZrN films. And we have added the detailed explanation of the SEM pictures. The revised manuscript has further confirmed the corrosion performance.

Round  2

Reviewer 1 Report

Now it can be accepted for publication.

Reviewer 2 Report

Authors have adressed the concerns. Article can be consider for publication.

Reviewer 3 Report

The authors properly addressed the comments, so the manuscript is suitable to be published in Materials.